# A Deep-Learning Based Pipeline for Estimating the Abundance and Size of Aquatic Organisms in an Unconstrained Underwater Environment from Continuously Captured Stereo Video

**DOI:** 10.3390/s23063311

**Published:** 2023-03-21

**Authors:** Gordon Böer, Joachim Paul Gröger, Sabah Badri-Höher, Boris Cisewski, Helge Renkewitz, Felix Mittermayer, Tobias Strickmann, Hauke Schramm

**Affiliations:** 1Institute of Applied Computer Science, Kiel University of Applied Sciences, 24149 Kiel, Germany; 2GEOMAR Helmholtz Centre for Ocean Research Kiel, 24148 Kiel, Germany; 3Institute of Communications Technology and Microelectronics, University of Applied Sciences, 24149 Kiel, Germany; 4Thünen Institute of Sea Fisheries, 27572 Bremerhaven, Germany; 5Fraunhofer IOSB, IOSB-AST Ilmenau, Fraunhofer Institute of Optronics, System Technologies and Image Exploitation, 98693 Ilmenau, Germany; 6Research Unit Marine Evolutionary Ecology, GEOMAR Helmholtz Centre for Ocean Research Kiel, 24148 Kiel, Germany; 7Department of Computer Science, Faculty of Engineering, Kiel University, 24118 Kiel, Germany

**Keywords:** marine species detection, underwater imagery, stereo-vision, deep learning

## Abstract

The utilization of stationary underwater cameras is a modern and well-adapted approach to provide a continuous and cost-effective long-term solution to monitor underwater habitats of particular interest. A common goal of such monitoring systems is to gain better insight into the dynamics and condition of populations of various marine organisms, such as migratory or commercially relevant fish taxa. This paper describes a complete processing pipeline to automatically determine the abundance, type and estimate the size of biological taxa from stereoscopic video data captured by the stereo camera of a stationary Underwater Fish Observatory (UFO). A calibration of the recording system was carried out in situ and, afterward, validated using the synchronously recorded sonar data. The video data were recorded continuously for nearly one year in the Kiel Fjord, an inlet of the Baltic Sea in northern Germany. It shows underwater organisms in their natural behavior, as passive low-light cameras were used instead of active lighting to dampen attraction effects and allow for the least invasive recording possible. The recorded raw data are pre-filtered by an adaptive background estimation to extract sequences with activity, which are then processed by a deep detection network, i.e., Yolov5. This provides the location and type of organisms detected in each video frame of both cameras, which are used to calculate stereo correspondences following a basic matching scheme. In a subsequent step, the size and distance of the depicted organisms are approximated using the corner coordinates of the matched bounding boxes. The Yolov5 model employed in this study was trained on a novel dataset comprising 73,144 images and 92,899 bounding box annotations for 10 categories of marine animals. The model achieved a mean detection accuracy of 92.4%, a mean average precision (mAP) of 94.8% and an F1 score of 93%.

## 1. Introduction

Marine ecosystems are very dynamic systems that change continuously under the influence of a wide variety of external factors. This is especially true for marine ecosystems, which—like the Baltic Sea—usually suffer from multiple pressures due to, e.g., fisheries, climate changes and discharges of pollutants and nutrients. The potential cumulative effects of overexploitation, on the one hand, and the simultaneous manifestation of anthropogenically influenced climate change, on the other hand, mean that commercially used fish stocks are the most endangered components of marine ecosystems. The underlying physical processes in the ocean occur on horizontal and vertical scales from millimeters to thousands of kilometers and on time scales from milliseconds to several decades and beyond [1]. To study these processes in sufficient detail, it is necessary to select survey design and investigation methods that allow the relevant scales to be spatially and temporally resolved. Thus, continuous and synchronous high-resolution multiparametric time series data makes long-term fixed-point observatories uniquely insightful in resolving decadal environmental trends required to understand the effects of global climate change.

In an ongoing effort of German universities and marine engineering companies, two Underwater Fish Observatories (UFOs) were developed to offer an alternative solution for the monitoring of fish populations and marine biodiversity. The UFO is a stationary, cabled sensor platform that is capable of continuously recording a variety of sensor data at high temporal resolution and continuously for several months. The sensors used include a low-light stereo camera, an imaging sonar, a CTD (conductivity, temperature, pressure) probe, an acoustic Doppler current profiler (ADCP), and a fluorometer. The UFO sensor platforms were deployed in different locations, i.e., at the seafloor level in the North Sea, close to the FINO3 research platform, about 45 nautical miles east of the island of Sylt, the Kiel Fjord as well as the Eckernförder Bight. The first prototype of the UFO, as deployed in the Kiel Fjord, and a map illustrating the deployment locations is illustrated in Figure 1.

During the several measurement campaigns, the primary goal was to conduct a continuous recording of stereo video and sonar data as well as various oceanic parameters at a high temporal sampling rate. In total, the raw dataset spans approximately 240 days, resulting in nearly 3000 h of video footage, as the cameras were not operated during the night hours. By using automatic pattern recognition (i.e., acoustic and optical fish detection, tracking and classification), the UFO is intended to allow for continuous all-season collection of data to assess a broad range of indicators. This is especially relevant to evaluate the status of various components of the Baltic Sea ecosystem with regard to achieving Good ecological and Environmental Status (GES) in the context of the Water Framework Directive (WFD) and Marine Strategy Framework Directive (MSFD).

The complete processing pipeline used to process the optical data recorded by the UFO, including the automatic determination of species, abundance and size of free-swimming underwater organisms, is described in this work. The use of digital imaging technology in marine research has become a widespread practice due to the rapid advancements in recent years regarding the available hardware and software. These advancements have led to digital cameras with improved image quality, increased storage capacity, and greater in situ applicability while also becoming more affordable in the consumer market. The application of modern sensors and algorithms for underwater imaging is diverse and has been the subject of numerous reviews, which have highlighted their general applicability in marine science [2,3,4,5,6,7].

The conventional standard practice of marine monitoring in fisheries science is to conduct classical trawl surveys with research vessels, which are based on more or less regular fish catches, in order to assess the dynamics, biomass, abundance and health of species populations aiming at managing these natural resources sustainably. This involves physically observing and collecting samples of marine organisms in the field and/or the manual interpretation of sonar recordings by a human observer. However, these methods may not be able to provide the same level of accuracy and detail as modern digital approaches. For example, manual surveys may be limited by the observer’s ability to identify organisms and may not be able to detect all species present in a given area. Specimen collection can also be problematic because it may not be possible to catch all individuals of a species, or it may have a negative impact on the population of the species. The weaknesses of these monitoring methods may be summarized as follows: these are invasive, cost-, labor- and time-intensive, discontinuous (snapshot-like) and asynchronous, highly aggregating, often lacking temporal and or spatial resolution, and are not allowed everywhere (restricted and protected areas, wind farms, etc.), not emission-free and not of low noise.

In contrast, new and innovative underwater technology can make monitoring and surveillance much more effective, less invasive, less expensive and more reliable, hence less prone to errors and misinterpretations. Such new technology will allow smart fish identification and quantification by developing autonomous technology solutions fully integrated into common underwater platforms. A core part of this is modern digital approaches, such as automatic computer vision algorithms, which can provide a more comprehensive and accurate assessment of the various parameters pertaining to marine biodiversity. These methods are able to analyze large amounts of data quickly and accurately, which allows for a more efficient and accurate assessment of marine biodiversity, as it reduces the need for manual analysis and increases the speed at which data can be collected and analyzed. Additionally, in the case of continuous observation, these methods can be used to detect changes in a marine habitat that are difficult for humans to recognize, such as subtle changes in the distribution of marine organisms.

The main contributions of this work are:The implementation of a processing pipeline for the automatic detection, abundance and size estimation of marine animals,A novel dataset, consisting of 73,144 images with 92,899 bounding boxes and 1198 track annotations for 10 different categories of marine species, was recorded in an unconstrained underwater scenario.

## 2. Related Work

Several deep-sea nodes have been deployed at key locations in world seas, from the Arctic to the Atlantic, through the Mediterranean to the Black Sea, e.g., MARS (Monterey Accelerated Research System) [8], NEPTUNE (North-East Pacific Time-Series Undersea Network experiments) [9] and VENUS (Victoria Experimental Network Under the Sea) [10], as well as shallow water nodes, such as COSYNA (Coastal Observing System for Northern and Arctic Seas) [11,12], OBSEA [13] and SmartBay (Smart Bay Cabled Observatory) [14]. For a more detailed overview of long-term observatories in operation worldwide, we refer to Zielinski et al. [15].

Regarding the related work in the field of fish detection using computer vision techniques, only those works are considered that deal with both the localization and classification of marine animals in underwater images, i.e., given a full image, the species and position of each visible animal is determined automatically and thereby also the class-specific abundance. Reports about single-image classification, such as those that use photographs or rectangular regions that have been cropped from full images and contain only one type of fish, are excluded since the main focus of this work is on the complete detection pipeline in natural underwater scenes. Therefore, the approaches considered relevant have been evaluated on image data taken in situ underwater rather than on a photo table or conveyor belt. Furthermore, only modern deep learning techniques are taken into account instead of more traditional object recognition algorithms, such as support vector machines (SVMs) making use of handcrafted features or shape-matching algorithms, such as the generalized Hough transform (GHT). Many of those excluded approaches have been the subject of the aforementioned reviews, and it has been expressed multiple times that they have been outperformed in terms of accuracy and robustness by more recent deep learning techniques. Until today, many different deep learning techniques have been widely adopted in the field of marine imaging due to their general applicability and superior performance compared to traditional methods. Advancements in deep learning, which were initially developed for seemingly unrelated fields, such as autonomous driving and medical imaging, have also been applied to marine imaging. This emphasizes the general applicability of this mostly data-driven family of algorithms, which is a huge advancement over traditional methods that often require specific domain knowledge and custom tailored solutions.

To the best of our knowledge, Li et al. [16] were the first to publish the utilization of a modern deep learning framework to localize and classify fishes in an unconstrained underwater environment in 2015, by employing a Fast R-CNN [17]. To train and evaluate the model, the authors used a custom version of the public dataset Fish4Knowledge [18], which contains 24,277 low-resolution images depicting 12 different fish species in a coral reef. It is worth mentioning that the authors create their own data-split and do not use the original training and test split of the dataset; however, they do not elaborate on the method by which the splits were created, such as random sampling or grouping by video sequence. Additionally, it is not explicitly mentioned on which portion of the data the presented results were obtained. In comparison to some alternative approaches, they found that the proposed method achieved the highest mean average precision (mAP) of 81.4%. In a subsequent publication [19], the authors integrated several advancements that had previously been introduced by Pvanet [20] and improved the mAP score to 89.95% on the same dataset.

For a comprehensive overview of the published attempts using deep learning to detect underwater organisms, it is recommended to refer to the reviews mentioned above. However, for the sake of completeness, several studies using YOLO are highlighted below, as this is the same model used in this study. Many well-established detection models are based on the YOLO framework, which was originally proposed by Redmon et al. [21] and has since undergone many improvements and variants [22,23,24,25]. A main property of YOLO is that it processes an entire image in a single forward pass, hence the acronym YOLO, which stands for You Only Look Once. By doing so, the processing speed is significantly higher when compared to other detection systems, however, at the cost of detection accuracy. Those alternative systems typically make use of multiple stages, e.g., a region proposal network followed by a classification module, as is used by Faster Region-Based Convolutional Neural Networks ((Faster R-CNN) [26]). Due to the widespread use of YOLO, especially in applications that have a high demand for real-time processing of image data, it can be considered a state-of-the-art detector. It is not surprising that YOLO, or one of its many variants, has already been applied in underwater imaging numerous times. This is due to the fact that underwater applications often have to be deployed on devices with limited computational resources, such as underwater vehicles or portable devices, or have to process large quantities of image data in a time-efficient manner, e.g., continuous videos, recorded by stationary cameras in areas of high population density or aquaculture settings.

Sung et al. [27] were the first to adopt YOLO for detecting free-swimming fishes. The study employed a relatively small dataset of 1912 images, which were sourced from a pre-existing dataset [28] and augmented with new images captured by an underwater vehicle. The authors reported that the system achieved 93% classification accuracy, with a 63.4% intersection over the union between the predicted bounding boxes and ground truth, at a frame rate of 16.7 fps. However, the model was only trained to detect any fish and was unable to differentiate between different fish species. In the study by Shi et al. [29], several deep learning-based object detection models were evaluated, including a modified version of SSD known as FFDet, which incorporates a separate feature fusion module, YOLOv3, Faster R-CNN and the original SSD [30]. The authors used the original train and test split of the Fish4Knowledge dataset to obtain the results and found that FFDet achieved superior performance with a mAP score of 62.8%. In the study conducted by Lu et al. [31], YOLO was utilized to identify four distinct aquatic species in deep-sea imagery, including shrimp, squid, crab and shark. The authors fine-tuned the model using a custom dataset comprising 10,728 images and found that YOLO yielded superior results in comparison to alternative methods. However, the authors did not provide detailed information regarding the configuration of the training and evaluation data. In other works, different variants of YOLO were used to detect sablefish Bonofiglio et al. [32], several nordic fish species [33], reef fishes [34,35] or fish in mangroves [36].

The automatic detection of fish is a prerequisite to measuring their size and estimating biomass. Several approaches for measuring the size of caught fish, such as those on a conveyor belt or photo table, have been published so far. However, those methods will not be further described since any photogrammetric measurement conducted underwater presents additional challenges compared to pure in-air measurements, e.g., due to the effects of refraction introduced by the camera housing, which can invalidate the single view-point assumption of basic perspective camera models [37]. Numerous investigations have been carried out on the effects of different parameters that affect the accuracy of size measurements performed underwater. For instance, Boutros et al. [38] investigated the effect of the calibration procedure, stereo baseline and camera orientation on measurement accuracy. Among other insights, they found that using a 3D calibration target, i.e., a wireframe cube, allowed for more precise measurements than calibration with a planar target, such as a checkerboard pattern. In Harvey et al. [39], stereo videos were utilized to measure free-swimming southern bluefin tuna. Specifically, the snout-to-fork length and the maximum body depth were measured with an average error of 1.72 and 1.37 mm, respectively. A drawback of the method was that the stereo images had to be manually synchronized using an optical trigger LED, and the selection of key points for the measurement was performed manually. The approach proposed by Garcia et al. [40] involves using iterative thinning and morphological operations on a previously determined semantic segmentation to estimate the length of a fish. However, this method does not take into account the 3D pose of the fish. A more advanced approach, as presented by Suo et al. [41], utilizes key points such as the eye, head and caudal fin to estimate the 3D pose of the fish in a stereo view. This allows for more accurate measurement of fish length, even when the fish is not swimming directly parallel to the camera. In their experimental results, the authors report the average estimation error for a fish length of 5.58%. Given the complexity and extensive applications of photogrammetric measurements in aquatic environments, interested readers are directed toward comprehensive reviews that cover various aspects of the topic, such as those presented in Jessop et al. [42], Shortis [43], Castillón et al. [44] and Cappo et al. [45].

In addition to the detection of fish, notable progress has also been made in the field of semantic segmentation of oceanic species, which involves predicting a specific outline rather than an enclosing rectangular bounding box. This type of segmentation is particularly useful for a more accurate estimation of animal sizes, as it provides the exact image area occupied by the animal. Various deep learning models have been applied successfully for free-swimming animals, such as cod and jellyfish, captured by stationary underwater cameras [46], up to 37 different marine animals, including those with camouflage, as part of the MAS3K dataset [47,48,49], fish passing through a camera chamber in a fish trawl [40] and multiple fish species in a tropical habitat [50].

## 3. Materials and Methods

The following sections will provide a description of the algorithms and practices used within this study. In particular, the following aspects will be discussed:The recording setup, including the information about the stereo camera and the recording conditions,The dataset used in this study, including the number of samples, the different categories of marine organisms included as well as the distribution of the samples among them,An overview of the complete processing pipeline,The object detection algorithm used to identify marine organisms in the images,The stereo camera system and its calibration process,The stereo-matching technique was applied to match detected organisms in the left and right images.

### 3.1. Recording Setup

A primary goal of the UFO is to be as non-invasive as possible to allow the recording of underwater organisms in their natural behavior. This requirement prevents the usage of active lighting, as it creates an undesirable attraction for fish [51]. Therefore, all stereo videos were captured using a custom-built stereo rig consisting of two Photonis “Nocturn XL” cameras. These cameras use a monochrome Lynx CMOS image sensor optimized for low-light scenarios and allow recordings without active lighting from dawn to dusk, which was verified to have a maximum water depth of 22 m in the North Sea. The stereo system has a baseline of 120 mm and is housed in a custom-designed underwater case with a flat viewport. The stereo camera is critical for the calculation of the distance and sizes of objects in real-world units. During the operation of the UFO, the two H.264-encoded camera streams are continuously recorded on external hard disks with an optical resolution of 1280 × 1024 pixels at a frame rate of up to 20 fps using a standard consumer PC.

### 3.2. Dataset

A curated subset of the complete optical data, which has been recorded using the UFO over different periods of time, was annotated for the development and evaluation of computer vision algorithms. Specifically, the annotated ground truth included in the dataset is suited for the development and evaluation of algorithms for three major computer vision tasks: object detection, classification and tracking. In a nutshell, the following annotation types are included:Bounding boxes: Each perceivable object of interest (OOI) is marked by a bounding box, as is common for most object detection tasks. The bounding box is defined by its (x, y) coordinates of the upper-left and bottom-right points spanning the rectangle.Classes: Each bounding box has a class label assigned to it, i.e., the animal’s taxa.Tracks: The bounding box annotations belonging to the same animal (object instance) are grouped into tracks and thus annotate the object’s movement over the course of a video sequence.Metadata: The metadata information for each image includes the geolocation, date and a timestamp for each recorded video frame, up to millisecond resolution.

To aid the annotation process, VIAME [52] was utilized, which enabled human observers to annotate each frame of a pre-selected sequence of underwater activity by considering the entire video. The ability to switch back and forth between the individual frames of a sequence during the annotation process was deemed an essential aspect, as it is frequently observed that obscured or poorly visible fish tend to be overlooked by human annotators when viewing individual still images alone. However, by observing the moving image, it becomes significantly easier for the human eye to detect and mark objects. As a result, all human observers were tasked with annotating each animal in each video frame in which it was visible using a bounding box, regardless of whether it was only partially visible. Additionally, the tracking of individual animals was limited to the duration of their visibility without making any assumptions about whether they had been previously observed in other sequences. It should be noted that the human annotators had no marine biology background. Thus they were trained beforehand on the appearance of the most prominent taxa found in the recording area. In all cases where the annotators were unsure of the species, they had to choose a general category, such as fish, jellyfish or simply an unidentified object. Each bounding box and track annotation was rechecked by a second observer and corrected as needed, such as for missing or incorrect annotations. In addition, annotations where the species of the animal depicted could not be determined with a high degree of certainty were later evaluated and, if possible, taxonomically categorized by marine biologists. In a common scenario where a single fish swims in and out of the image area, the annotations may include samples of just the head or the fin, the whole fish, as well as any other part of the fish that the human observer could make out. Additionally, the overall image quality is affected by the dynamic lighting conditions during the day or by weather conditions, as well as by water turbidity, which can be degraded by the amount of suspended solids and dissolved organic matter (DOM) in interaction with the water current. In any case, as long as a human annotator is able to identify and mark an aquatic organism using the full information from the video, the image will be included in the dataset. This dataset reflects the human ability to solve the task at hand, and we consider this to be the larger goal to be achieved or overcome by an automated system.

The data used in this study includes image samples for 10 different categories of fish or jellyfish taxa commonly found in the western parts of the Baltic Sea. Whenever possible, each sample was assigned to a specific species. However, this was not always possible, for example, in cases of poor visibility or when an animal was very close to or far away from the camera. In situations where it was not possible to determine the exact species, but it was still obvious that the animal was either some kind of fish or a jellyfish, those parent object classes were used with an additional *unspecified* marker. Additionally, in some cases, the taxonomic family was chosen instead of the specific species, e.g., as it is difficult to differentiate between a sprat (*Sprattus sprattus*) and a herring (*Clupea harengus*) relying only on visual cues, in which case the parent family *Clupeidae* was chosen. The amount of annotations for each category is listed in Table 1.

It is apparent that the dataset is not evenly distributed in terms of the number of annotations per class. This is also evident from the illustration in Figure 2. Approximately 75% of the data consists of samples from various species of jellyfish, with *Aurelia aurita* and *Ctenophora* being the most prevalent classes. The abundance of jellyfish samples can be attributed to the fact that they can be annotated semi-automatically with greater reliability than fish. Due to their slow and linear movements, the annotation process could be assisted by the interpolation tool provided by VIAME, with minimal need for manual adjustments. In brief, it is possible to annotate only the start and end points of a tracklet (a segment of a full track), and the tool can interpolate the position and dimension of bounding boxes in between those endpoints. This is, however, more challenging to accomplish for swimming fish, as they can rapidly change their swimming speed and direction from one video frame to the next.

Real-world datasets, especially those with video recordings of animals in their natural habitat, are often plagued by significant class imbalance problems [18,53]. The impact of such skewed data distributions on the training of deep learning models is a topic of current research, and numerous strategies have been proposed to mitigate the detrimental effects [54]. To increase the heterogeneity of our dataset during the experimental phase, we employed several conventional techniques, including the introduction of image augmentations, such as random scaling, translation and brightness. However, given the complex nature of the class imbalance, it is clear that further research is needed to find more sophisticated and effective methods to address this challenge in the context of marine underwater species detection. 

### 3.3. Processing Pipeline Overview

The proposed processing pipeline, as illustrated in Figure 3, is a multi-step process that utilizes a combination of techniques to detect and measure marine organisms in underwater stereo camera streams. The first step of the pipeline involves recording and storing the stereo camera streams on a hard disk for later processing. This step ensures that the raw data are preserved for further processing and analysis.

An initial filtering step is applied using an adaptive background estimation and foreground detection method to efficiently process the large volume of raw data. The goal of this step is to identify sequences in the raw video streams that contain any kind of activity, thus reducing the amount of data that needs to be processed in the next step. This approach allows for faster roundtrip times when testing different settings or models on the same dataset. However, it should be noted that in an online processing scenario, this step is expected to be unnecessary as the subsequent detection module is much more efficient in terms of detection accuracy and runtime.

The next step in the pipeline is the detection module, which utilizes a Yolov5 model that has been trained to detect marine organisms. This model is applied to each frame of the previously identified activity sequences to detect each animal of interest. The detection module returns the detected animals for each frame in the form of enclosing bounding boxes, along with the predicted animal species.

The final step of the pipeline is the stereo processing module, which is responsible for calculating the size of each animal and its distance to the camera. This step is based on a matching scheme between the detected bounding boxes on each of the synchronized stereo frames. The module calculates the size and distance of each animal using the information obtained from the stereo camera, which allows for more precise measurements. However, it should be noted that stereo matching may not always be possible, which will be discussed later.

### 3.4. Activity Detection

The initial videos are pre-processed using an adaptive background estimation algorithm, as outlined in Zivkovic [55], to identify sequences that display any form of movement. In summary, this algorithm compares each frame of the video to a background model, which is initially built from a configurable amount of video frames and is updated over time as new frames are processed. Any pixel values that deviate from the background model are considered part of the foreground, which, in this case, would be any movement or activity in the video.

The background is modeled by a Gaussian mixture model (GMM), where each pixel in the video is represented by a set of pixel values, and a GMM is fit to the distribution of these values over time. When comparing each new frame to the background model, any pixels whose values deviate significantly from the Gaussian distributions are considered part of the foreground.

Within this work, the individual frames are additionally median filtered to eliminate particle noise and make the background estimation less sensitive to minor changes in the image. The objective of this activity detection is not to produce an accurate foreground mask but rather to detect significant changes from one video frame to the next, thereby reducing the vast amount of video data to only those segments that may contain an animal. In the subsequent detection module, those activity sequences will be further processed to detect the objects of interest. The activity detection module was used at the beginning of the project before a trained model such as Yolov5 became available. Using this conventional method, which does not require training data, it was possible to reduce a large amount of video data to more relevant sequences and identify potential candidates for annotation. Numerous video sequences from different recording days were manually validated to ensure the accuracy of the activity detection results. The recordings were made under varying light conditions, ranging from sunrise to sunset, and in a diverse range of visibility conditions, from relatively clear and calm water to heavily turbid water due to the presence of suspended matter and strong water currents. It was found that no activities involving animals were missed, but on the contrary, sequences were extracted that showed activities of no particular interest but also barely visible fish that were missed by a human observer at first glance. The main purpose of the activity detection system is to serve as an initial filter to reduce the amount of raw video data to a manageable level and analyze it more effectively. Therefore, we did not calculate specific metrics to evaluate its performance. However, based on our experience, it can be said that the system exhibits a high recall rate but a relatively low precision rate. Nevertheless, for our specific use case, this level of performance is considered adequate, as the primary workload is taken on by the subsequent detection module.

In a real-time detection system, such as that used on an underwater vehicle, this aspect of the processing pipeline would most certainly not be required, as the trained detection module can now detect the objects of interest with increased robustness and reject the majority of unwanted objects.

### 3.5. Detection of Marine Species

To automatically localize animals that are captured by the camera and determine their species, the deep learning model Yolov5 [56] is utilized, which builds on the previous YOLO version [24] and improves it in several ways. One of the main advantages of this newer version is that it allows for variable input sizes and comes with pre-trained weights for different configurations. This flexibility allows the trade-off between model complexity and recognition accuracy to be adjusted. The ability to adjust the model size is particularly useful for this study, as it is planned to deploy the detection system on a mobile underwater vehicle with limited resources. Therefore, the developed algorithms and tools can be used for offline processing scenarios as well as real-time processing.

YOLO is trained using a set of images in which the objects of interest are labeled with bounding boxes and object categories, in this case, animal species. The training process of YOLO is formulated as a regression problem, where the goal is to predict the coordinates of the bounding box and the object class for each object in the image. In this problem, the network takes an image as input and produces a set of predictions for the bounding boxes and object classes for that image, represented by their coordinates and class labels. The network is trained to minimize the difference between the predicted bounding boxes and their corresponding human-annotated ground truth, as well as the difference between the predicted and target class labels. The annotation protocol used to determine the ground truth has been described in Section 3.2.

In addition, YOLO uses an anchor box mechanism, which is a predefined set of bounding boxes with different aspect ratios and scales. Instead of concrete coordinates, the network predicts the relative offsets between the anchor boxes and the ground truth boxes, thereby reducing the number of parameters to learn. An improvement in Yolov5 over its predecessors is that these anchors are determined automatically, based on the bounding box sizes present in the training set, whereas previously, they had to be chosen manually.

Each detection is assigned a confidence score, which is a combination of the objectness and classification score and indicates the likelihood that the detected animal is a member of the target species. The confidence values can be used to filter out detections that are unlikely to be correct, improving the overall accuracy and reliability of the detection results. The impact of the confidence value on the overall performance is part of the experimental evaluation presented in Section 4, particularly visible in the F1 and precision-recall curves presented therein. Additionally, a class-agnostic non-maximum suppression (NMS) was used to eliminate duplicate detections. Specifically, any detections with an intersection over union (IoU) greater than 0.6 were discarded, and only the detection with the highest confidence score was retained in case of a match.

Yolov5 offers a range of data augmentation; vertical or horizontal flipping, rotation or translation. Additionally, it introduces a new technique called mosaic augmentation, which merges cutouts of four different images from the training set together into a new image and thereby moves the target object into a new image context. A detailed evaluation of the mosaic augmentation technique is given in the original publication for Yolov4 [24], where it was first introduced.

This technique is typically utilized in scenarios involving complex backgrounds, where it can help to improve the robustness of the model. However, given the more homogenous nature of the background in the used underwater dataset, the mosaic augmentation technique was not utilized in the final experiments. This decision was made to avoid the potentially detrimental effects of the rather fierce random cropping that occurs during the augmentation process, which can lead to undesired effects, such as the loss of important target object information. For instance, during the evaluation, it has been noticed that in several cases, not only the entire fish was detected, but also individual body parts such as the head or tail, resulting in multiple bounding boxes for the same fish. This behavior could possibly be explained by the mosaic augmentation technique, which may introduce unintended crops in the dataset. However, it is important to note that further ablation studies are necessary to confirm this explanation.

### 3.6. Stereo Processing

In order to perform stereo measurements, it is essential to first calibrate the cameras, which was carried out in accordance with well-adopted procedures. Specifically, a checkerboard pattern with a field size of 9 × 8 squares and a square size of 50 mm was employed as the calibration target, as illustrated in Figure 4. The target was recorded by the stereo cameras in various positions, orientations and distances relative to the stereo camera system. To ensure the most precise measurements, a diver positioned the checkerboard target before the cameras after they were deployed underwater. This is crucial, as the optical properties of images recorded underwater differ significantly from those recorded in air since the water itself acts as a medium that distorts the light, caused by factors such as refraction, absorption and scattering of light. By calibrating the cameras in situ, these distortions and variations can be accounted for to a suitable degree, providing more accurate and precise stereo measurements [43,57]. In this study, the camera calibration toolbox for Matlab [58] was used to calculate the intrinsic and extrinsic camera parameters. This implementation follows the calibration procedure, as has been described by Heikkila and Silvén [59], where a calibration pattern is used to compute correction coefficients for the radial and tangential distortions introduced by the camera lens, which are needed to extend the idealized perspective camera model.

Still, the task of calibrating underwater images poses additional challenges due to the numerous and complex factors that affect image formation, particularly the effects of refraction that are introduced by the camera housing. The light entering the housing must pass through water, then glass and finally air, resulting in double refraction, which must be accounted for. Moreover, there are other factors that can impede the calibration process, including light absorption, scattering by water and suspended particles, and poor visibility caused by high turbidity. As a result, the precision and accuracy of underwater calibrations are expected to be lower compared to in-air calibration, leading to potential scale errors in measurements [43]. Despite the considerable challenges, researchers have been exploring various strategies to tackle these issues, such as utilizing specially designed dome ports to eliminate refraction effects and fulfill the single-view point assumption of the perspective camera model [60] or explicitly modeling refraction effects on flat view ports to improve calibration accuracy [37].

To this end, we solely rely on the calibration parameters to absorb the described effects of refraction. It should be noted that there will always be a systematic error in measurements acquired through this approach since the extremal light rays do not intersect in the common center of projection, rendering the perspective camera model invalid [37]. However, the systematic error is generally less significant than the precision with which the measurements can be extracted [43].

In order to minimize measurement errors, meticulous attention was given to the calibration process. Specifically, the calibration was executed in situ at the observation site, thereby minimizing the potential impact of camera misalignment that may arise during the deployment of the UFO. This approach also accounts for the unique water properties present at the site, including salinity, which can influence the refractive index of water. To improve the precision of the calibration process, multiple images were acquired at varying distances up to the maximum viewing distance of the cameras. While capturing the calibration target from different angles was necessary, performing the calibration at all recording distances was imperative to minimize errors since measurement errors are most pronounced when measurements are taken outside the calibration range [43].

In total, 94 image stereo pairs, approximately recorded at a distance of 0.5–2.0 m, were selected to perform the calibration. The intrinsic and extrinsic camera parameters, as yielded by the calibration procedure, can be used to triangulate the 3D coordinates of any image point seen in both cameras at the same time and ultimately calculate the distance between any of those points, e.g., to measure the dimensions of a fish. The calibration procedure resulted in a mean reprojection error of 0.85 pixels, which measures how well the calibrated camera can reproject 3D points onto 2D image coordinates. While a low reprojection error is an important prerequisite for accurate photogrammetric measurements, it does not guarantee accurate size measurements of objects in the scene. The accuracy of the size measurements depends on various additional factors, such as the quality of the images, the distance and orientation of the object from the camera, and the processing algorithms used to extract the object dimensions from the images.

Ensuring the validity of calculated distances and positions is crucial for obtaining reliable and trustworthy results. While a fixed calibration target with precisely known dimensions and distances to the camera system is an ideal validation method, such a target was not available at the deployment site. Consequently, we utilized the simultaneously recorded sonar data to perform the validation. The camera and sonar data were recorded in a timely synchronized manner, enabling the mapping of the recordings to each other using their respective timestamps. This mapping facilitated the validation of the distance and horizontal extent of an object visible in both recording systems by comparing the calculated 3D information to those directly measured by the sonar. It is important to note that this validation was performed manually on several selected events, as automatic mapping of objects detected in both systems has not yet been realized. In Figure 5, an example of a validated stereo measurement is provided. In the camera image, a diver can be seen holding a calibration target, which is visible in both the sonar and camera views. It is evident that the estimated distance aligns with the distance measured by the sonar.

Besides validating the calculated distance of an object to the camera system using the sonar, additional efforts were undertaken to validate the size measurements. The sonar can measure the horizontal extent and distance of objects, but not the vertical dimension. Furthermore, the minimum distance that can effectively be resolved with the sonar is greater than the minimum camera range. Therefore, the stereo camera measurements were further validated using the calibration target described above. This target allows for automated 3D calculation, as the corners of the checkerboard field can be reliably detected automatically. By using the known size and structure of the calibration pattern, the distance between two given points of the checkerboard can be calculated and compared to the true distance in real-world units. This validation provides better insight into the performance of the stereo system regarding size measurements of objects, as opposed to simply looking at the reprojection error. The complete procedure to validate the size measurements involves the following steps:For a given set of timely synchronized stereo image pairs, detect the inner corners of the checkerboard in each image.For each detected corner, triangulate the 3D position using the previously calibrated camera parameters and the pixel positions of the corner in both images.For each corner, calculate the 3D distance to all points that lie on the horizontal and vertical lines of the checkerboard.Given the known checkerboard dimensions, calculate the error between the measured and expected distances, i.e., the number of checkerboard fields times the real-world size of a single square.

Subsequently, the horizontal and vertical measurement errors can be estimated based on the distance between two points and on a per-image level, for instance, by analyzing the average error over all point pairs per image. By following the aforementioned procedure and considering all 94 calibration image pairs, an overall average horizontal measurement error of 3.6 mm and a vertical measurement error of 5.1 mm could be observed. The dependency of the vertical and horizontal measurement errors on the recording distance is illustrated in Figure 6, which shows an increasing trend of the error with rising distance. A likely explanation for this is the gradually declining visibility of the target pattern, which makes it more difficult to precisely detect the needed target points, as well as the limited camera resolution.

While any image point seen in both camera views can be used to calculate 3D information, the main difficulty is to perform the matching of said image points to each other automatically, which is also known as the stereo correspondence problem. To this end, the proposed system uses a matching strategy based on the located bounding boxes and their respective geometric properties.

The initial step in the process utilizes the intrinsic parameters to correct distortions specific to each camera. Subsequently, the extrinsic parameters are employed to rectify the stereo images, thereby enabling the utilization of epipolar geometry methods [61], which constrains the search for corresponding image points to a horizontal line in both images.

To establish correspondence between the bounding boxes, detected in each stereo image, a matching score $m_{score}$ is calculated by considering both the overlap iou between the boxes, as determined by the intersection over union (IoU) metric, and the angle θ between the horizontal axis and the line connecting the center points of the two boxes. However, any match is always rejected if the predicted species for the two boxes do not correspond, if θ is larger than a configurable threshold $\theta_{thresh}$ or iou is below a configurable threshold $iou_{thresh}$.
(1)$$m_{score}\left(box_{1},box_{2}\right)=\left\{\begin{array}{cc} 0 & \mathrm{if}\;\theta >\theta_{thresh},\;iou<iou_{thresh},\;box1_{cls}\neq box2_{cls}\\ \frac{iou}{\theta } & \mathrm{otherwise} \end{array}\right.$$
with
$$iou=\frac{area(box_{1}\cap box_{2})}{area(box_{1}\cup box_{2})}$$
$$\theta =\arctan\left(\frac{y_{2}-y_{1}}{x_{2}-x_{1}}\right)$$
where $box_{1}$ and $box_{2}$ are the bounding boxes in the left and right images, $\left(x_{1},y_{1}\right)$ and $\left(x_{2},y_{2}\right)$ are the center points, and $box1_{cls}$ and $box2_{cls}$ are the classes predicted for those boxes, respectively. If many fishes have been detected on the same epipolar line, it may be possible that a single bounding box is matched to multiple other bounding boxes; in this case, the established match is chosen based on the highest matching score.

In summary, the matching of two bounding boxes involves checking (1) the angle of the horizontal line connecting the center points, (2) the IoU between the bounding boxes and (3) the species assigned to each of it. Finally, the matched bounding boxes are used to calculate the distance and size of the detected animal. The images depicted in Figure 7 show two exemplary results of the proposed stereo matching approach: an easier task, where only one jellyfish has been matched, and a more challenging one, where individual fish in a large swarm had been matched.

### 3.7. Abundance Estimation

The abundance of each detected species is determined based on the previously described activity sequences, i.e., sequences that are characterized by uninterrupted animal activity. Using these sequences, a metric, referred to as MaxN [62], will be used as a sampling regime to determine the relative abundance of animals that have been observed. This method involves counting the maximum number of individuals of a given species that can be observed within a defined observation area at any one time. The estimated abundances are then aggregated over configurable time intervals, such as per hour or per day, and can later be used to estimate the total population size, taking into account factors such as spatial and temporal variation in the distribution of the animals. Since the abundances are calculated based on activity sequences, this implies that individual animals re-entering the recorded area in a subsequent activity sequence time interval will be counted as a new individual. In other words, to this end, with the described methods, it is not yet possible to determine if an individual has been seen before or not. As a consequence, when utilizing this system for population sampling or conducting a census, a number of crucial factors must be considered to effectively address issues related to over- and undersampling.

These factors encompass a range of considerations, including the observation area, target species and sampling design, among others. Given the diversity of these factors, there is no one-size-fits-all solution to accurately estimate animal abundance in different contexts. Instead, a multitude of statistical methods and survey designs are available, each with its strengths and limitations. These factors interact in complex ways and require a thorough understanding of both the statistical and ecological principles underlying the different methods available. In light of the nuanced nature of these considerations, it is beyond the scope of this publication to delve deeply into the statistical methods and survey designs applicable to estimating animal abundance. Instead, the reader is referred to the relevant literature, such as the work published by Gilbert et al. [63], Moeller et al. [64] and Denes et al. [65].

## 4. Experimental Results

The dataset described above was partitioned into a training and validation set using a ratio of 0.85, achieved by random sampling. As is typical, the training set was employed to calculate the model loss and perform weight updates, while the validation set was utilized to monitor the training procedure, for instance, to detect overfitting and to control early stopping. It is noteworthy that the data split was executed on a per-image basis and not per bounding box sample, which guarantees that all annotations for a given image are included in only one of the subsets. However, given that the dataset consists of image sequences from a video with a relatively high frame rate, it is likely that subsequent images from that sequence can look quite similar. This is particularly true for sequences that depict slow-moving objects, such as different jellyfish. Therefore, the random sampling employed can place very similar images in the data-splits, which may lead to an underestimation of the error. In our experience, it is beneficial to make the validation set somewhat more challenging, as otherwise, early stopping may occur too soon, and the validation results may appear too optimistic in general. Therefore, additional measures were taken to include several examples in the validation set that were recorded on days for which not a single sample was included in the training set. These more difficult samples account for 18.5% of the images and 15% of the bounding boxes included in the validation set. In addition to the manually annotated images, 8441 images without marine organisms were included in the training set. Adding such background images is generally said to help Yolov5 to reduce the number of false positive detections [56], which we can agree with from our own experience. In our case, these images include a variety of backgrounds, such as water and sand, as well as several objects that should not be detected at all, such as sediment, plant parts or the wiper used to regularly clean the camera lens. The precise numbers of the samples per object category in the training and test sets are listed in Table 2.

For the presented results, we utilized the available pretrained weights for the Yolov5l6 model configuration and performed transfer learning on the described training data. We also compared the results with a model that was trained from scratch. However, this had no significant impact on the final performance and only resulted in a longer convergence time. The model was trained utilizing four NVidia Titan RTX graphics processing units, with a batch size of 32 and the stochastic gradient descent optimization algorithm [66], with a patience parameter of 25. Additional hyperparameter settings employed during training are detailed in Table A1, Appendix A. Despite reaching the default upper limit of 200 epochs of the used YOLO implementation, the training did not converge. Consequently, it was continued from the best checkpoint for an additional 129 epochs until no further improvement was observed and at which point the training was terminated in accordance with the established patience parameter. Assuming identical hardware configuration, the inference time required to process the image data with the selected model size is estimated to be around 20 ms per image. However, the processing time is further impacted by the post-processing procedures, particularly the non-maximum suppression (NMS). The duration of the NMS may fluctuate considerably, ranging from 1 to 20 ms, depending on the number of detected objects.

The results obtained on the validation data were evaluated using common metrics in the field of object detection, including the F1 score, the mean average precision (mAP) and a confusion matrix.

The mAP provides an overall measure of the model’s performance by taking the mean of the average precision (AP) over all object categories *N*. The AP is calculated by averaging the precision of the model at different recall levels. Specifically, we use the mAP@0.5 score, which accepts a prediction as true positive, if the IoU with the respective bounding box is at least 0.5.
(2)$$mAP=\frac{1}{N}\sum_{i=1}^{N}{\left(AP\right)}_{i}$$

The F1 score is a measure of the trade-off between precision and recall, and is calculated as the harmonic mean of precision and recall. Mathematically, it is expressed as:(3)$$F1=2*\frac{precision*recall}{precision+recall}$$

Given a confusion matrix, the rows represent the true class labels, while the columns represent the predicted class labels. The diagonal elements represent the number of observations that have been correctly classified, and the off-diagonal elements represent the misclassifications. Furthermore, the confusion matrix illustrated in Figure 8 also includes a background class, enabling a more thorough evaluation of performance for each individual object class. Specifically, the background row includes instances where an object class was inaccurately detected as background or false negatives. Conversely, the background column depicts the proportion of each object class among all false positive detections or instances where the class was identified; however, in reality, only background was present.

Upon examination of the confusion matrix, it is evident that several categories exhibit nearly optimal performance with an accuracy of greater than 97%, specifically *Aurelia aurita*, *Ctenophora*, *Gadus morhua*, *Scomber scombrus*, *Pleuronectoidei* and *Jellyfish unspecified*. Conversely, the model demonstrated the lowest performance for *Salmonidae* and *Cyanea capillata*, with an accuracy of 81%. Subsequently, the model performed slightly better for *Fish unspecified* and *Clupeidae*, achieving 84% and 85% accuracy, respectively. Overall, the mean detection accuracy across all 10 object categories on the validation data amounts to 92.4%.

The F1 plots and Precision-Recall curves illustrated in Figure 9 show that the maximum mean F1 score of 93% was achieved at a confidence level of 0.625 and the average mAP at an accepted IoU 0f 0.5 amounts to 94.8%

Some further understanding can be gained regarding the nature of errors made by the model. For instance, 7% of errors made for *Cyanea capillata* are the result of misclassifying it as *Aurelia aurita*. This can be easily understood, as both types of jellyfish can appear similar, particularly under conditions of poor visibility. A similar scenario is observed between *Clupeidae* and *Fish unspecified*, with 8% and 6% of cases being mistaken for each other, respectively. The *Fish unspecified* class is particularly difficult to detect due to its ambiguity and wide range of variations, from poorly visible fish in the background to small, blurry fish in the foreground. This also explains the high rate of false positives (33%) for this class, as different types of poorly visible objects may be mistaken as fish. Additionally, a group of *Clupeidae* may contain some fish that look like *Fish unspecified*, e.g., if swimming in the background, but were still annotated as *Clupeidae*. Surprisingly, the *Jellyfish unspecified* class does not exhibit this type of problem, though it should be noted that their absolute number in the dataset is much smaller compared to that of *Fish unspecified*. Several exemplary cases of faulty detections are illustrated in Figure 10.

## 5. Discussion

The use of deep learning models, such as Yolov5, to detect marine species in images has shown promising results. However, as with any machine learning model, there are limitations and challenges that still need to be addressed to improve the accuracy of detection.

One of these challenges is underrepresented classes in the training data, such as the taxa of *Salmonidae*. Given the relatively small amount of training data for this class, the model may not be able to learn the distinct visual features of these taxa, resulting in incorrect detections. To address this, it is recommended to increase the amount of training data for these underrepresented classes. Additionally, using a hierarchical loss, as in Gupta et al. [33], can penalize the model based on the taxonomic hierarchy. Given this, the detection accuracy could be further improved, especially regarding misclassification between different taxonomic branches, e.g., *Salmonidae* and *Gadidae*.

Another issue is the confusion between similar classes, such as *Clupeidae* and *Fish unspecified*. In this case, it is difficult to define a precise boundary between these classes, which may be difficult to resolve solely through the detection system. To address this, using tracking algorithms, such as the one described in [67], can help to classify the fish more accurately, especially in frames where the fish is partially obscured. However, although considerable progress has been made in the field of multi-object tracking, it is still challenging to detect multiple objects simultaneously, especially if they are visually similar, as is the case in swarms of fish. The tracking may also be computationally intensive and, therefore, hard to deploy on resource-limited hardware.

The presented method for the automatic size estimation of underwater organisms is based on straightforward stereo matching, which relies primarily on a properly calibrated stereo setup and the geometric properties of the bounding boxes as predicted by the detection system with high accuracy. While this method exhibits notable simplicity, enabling efficient implementation even on hardware with limited capabilities, its measurement accuracy is limited in certain respects. The proposed method performs reasonably well if the detected individuals do not appear on the same epipolar line or otherwise if they are of different taxa or have different sizes and orientations, which would be reflected in different geometric properties of the bounding boxes. However, to improve the performance of this method, image features, such as feature vectors extracted by a CNN, can be used in conjunction with a distance function, such as cosine similarity. Another alternative is to also incorporate morphological characteristics, such as the fish contour or specific key points, as has been performed by previous research [41]. These methods would likely improve the stereo correspondence problem in more challenging cases but at the cost of increased computational complexity, which may not always be feasible, especially when working with resource-limited hardware.

The MaxN approach, used to estimate the abundance, results in more conservative estimates, particularly when large numbers of animals are observed over an extended period of activity, as reported in the literature [68]. For example, if a large group of fish is passing the camera, the MaxN approach would provide the maximum number of fish detected in the same video frame. However, a portion of the fish group could have already passed the camera or may still come, and those individuals would not be included in that count. An alternative method includes using an average count of the number of fish [69], which, however, is prone to overestimate the abundance since the same fish may be counted multiple times during the activity sequence. An even more sophisticated alternative would be to use an additional object tracking method, i.e., to identify the trajectory of a detected animal over the course of the image sequence. The thereby detected tracks would allow determining exactly when an individual animal is entering or leaving the field of perception, which could be used to reduce the effect of statistical measurement by utilizing the number of tracks instead of the number of detections per time frame. An additional possibility to improve the abundance estimation is to use other sensors, specifically the multibeam sonar system, which is installed on the UFO. It provides the ability to detect any movement over a very large area, especially if it is not just a single fish but a large school of fish moving within the area imaged by the sonar. This covers a much larger observation area than the cameras. In a sensor fusion approach, the animals detected and classified by the camera could be mapped to the detections in the sonar view and be tracked over a much larger area, which would reduce the effect of counting the same individual repeatedly. When an individual fish leaves the field of view (FOV) of the camera, it could be tracked using the sonar system. When it re-enters the FOV of the camera, it could be identified as the same fish, thus avoiding double counting of the same individual.

## 6. Conclusions and Outlook

In conclusion, this paper presents a comprehensive processing pipeline for the automatic detection, localization and characterization of biological taxa in stereoscopic video data. Within this pipeline, the raw data are pre-filtered by an adaptive background estimation and further processed using Yolov5 to extract the location and type of 10 different taxa classes in each video frame, which is then used to calculate stereo correspondences and approximate sizes of the depicted organisms. This pipeline provides a valuable tool for the study and monitoring of marine biodiversity and conservation.

While the process of identifying several marine animals based on stereo camera images in the water is challenging enough, the next step will be to transfer and adapt those algorithms to a mobile system, e.g., an underwater vehicle. In the current project, one partner developed such a vehicle, as depicted in Figure 11, that can be operated remotely (as a ROV—Remotely Operated Vehicle) or autonomously (as an AUV—Autonomous Underwater Vehicle). The specifications of the vehicle were set by all partners, and they include weight, diving depth, payload sensors and expandability. The most important specification was the fact that the same sensors as for the stationary setups should be used, so the stereo camera and the sonar are identical to those. As can be seen in the images, both sensors can be found in the front of the vehicle. In addition to that, a powerful computer has been integrated to process the sensor data in real-time. The advantage of having a mobile device is clearly an extended operational area with more object categories. However, the challenges for this mobile setup increase with the extended range. There will be more object motion in the images, no static background, much more variable lighting and also more biological taxa to be labeled and trained.

There is a growing need for long-term ocean observation, as coastal and deep ocean observatories are key to improving the understanding of the underlying processes within the marine ecosystem [70]. This is the reason why different European initiatives have promoted their use through investments in infrastructure development and dedicated scientific sessions on how to coordinate ocean data acquisition, analysis and dissemination, e.g., JERICO S3 (Joint European Research Infrastructure of Coastal Observatories: Science, Service, Sustainability). Among those initiatives, the European Multidisciplinary Seafloor and Water-Column Observatory (EMSO) is a pan-European distributed research infrastructure, which has the major scientific objective of long-term monitoring, mainly in real-time, of environmental processes related to the interaction between the geosphere, biosphere and hydrosphere, including natural hazards.

Our goal with this research is to propose an additional solution that can have a positive impact on the monitoring of oceanic ecosystems. We believe that by leveraging the latest advancements in technology and scientific knowledge, we can play an important role in understanding and protecting the oceans that sustain us.

## Figures and Tables

**Figure 1 sensors-23-03311-f001:**
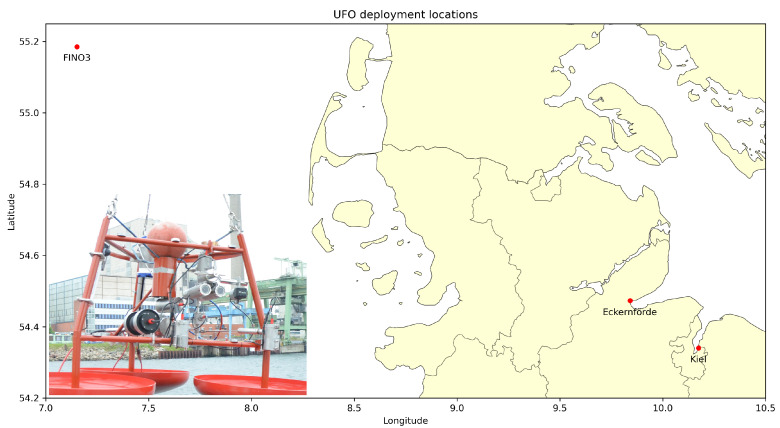
The prototype of the Underwater Fish Observatory (UFO), shortly before deployment in the Kiel Fjord, and a map displaying the platform’s deployment locations to date.

**Figure 2 sensors-23-03311-f002:**
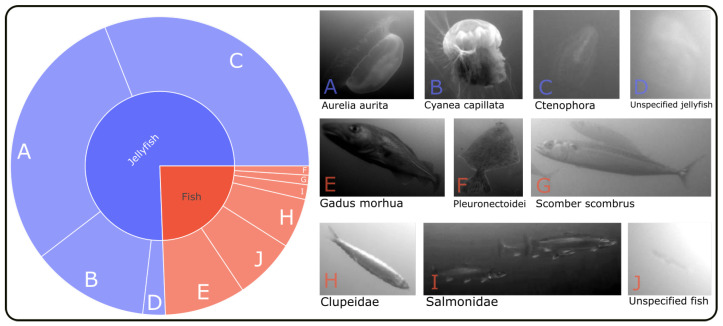
The distribution of classes among the 92,899 bounding box annotations in the used dataset. Included are ten different object categories, with roughly 75% of the samples consisting of four different types of jellyfish and the remaining 25% falling into six categories of fish.

**Figure 3 sensors-23-03311-f003:**
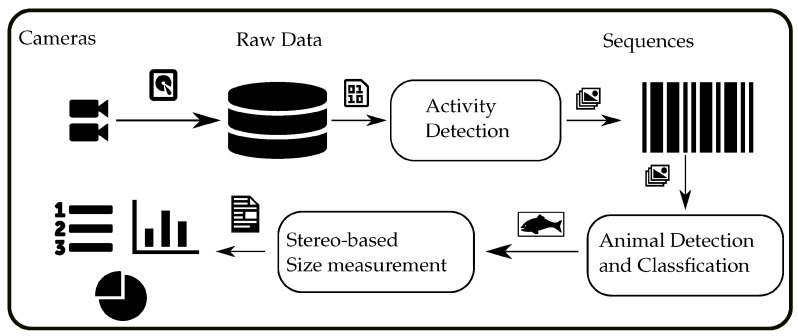
The processing pipeline from the recording of underwater videos to the final biomass estimation.

**Figure 4 sensors-23-03311-f004:**
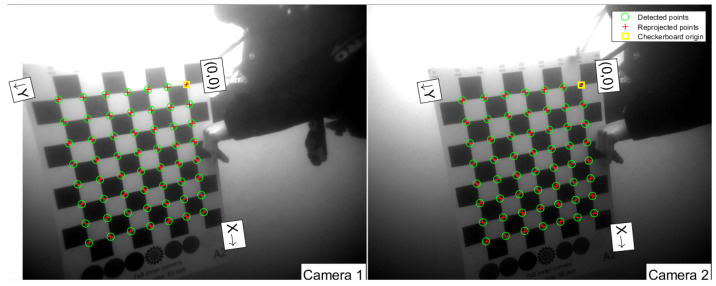
Example of an image pair used for the calibration with the Matlab camera calibration toolbox [58].

**Figure 5 sensors-23-03311-f005:**
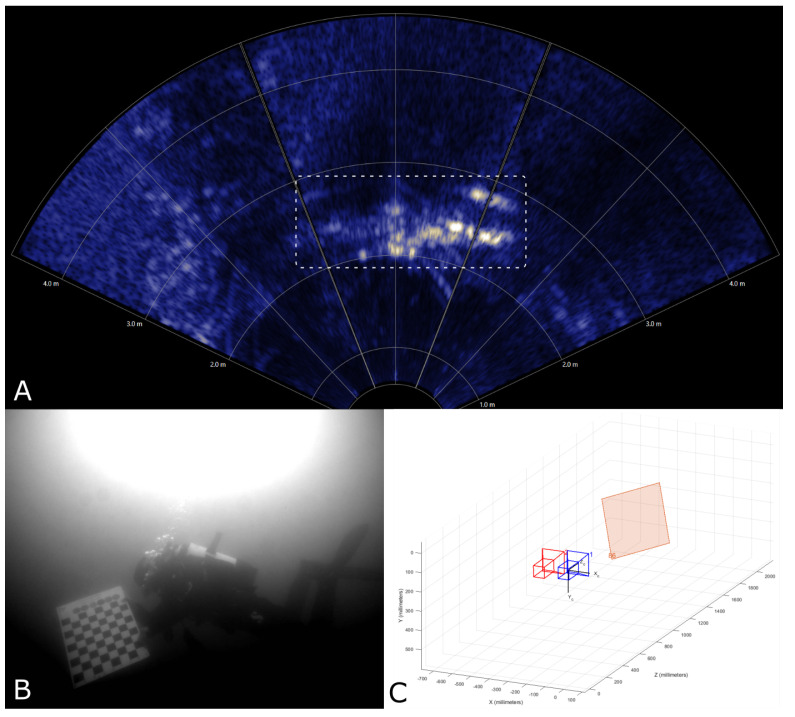
To validate the stereoscopic measurements, the synchronized sonar data were used. Here, the sonar (**A**) measures the distance of a diver holding a calibration target (marked with a white rectangle) at ∼2 m. Using the camera data (**B**), the second camera view has been omitted here, the 3D position of the same calibration target, relative to the two cameras (represented here in red and blue), has been estimated at a distance of ∼2 m as well (**C**).

**Figure 6 sensors-23-03311-f006:**
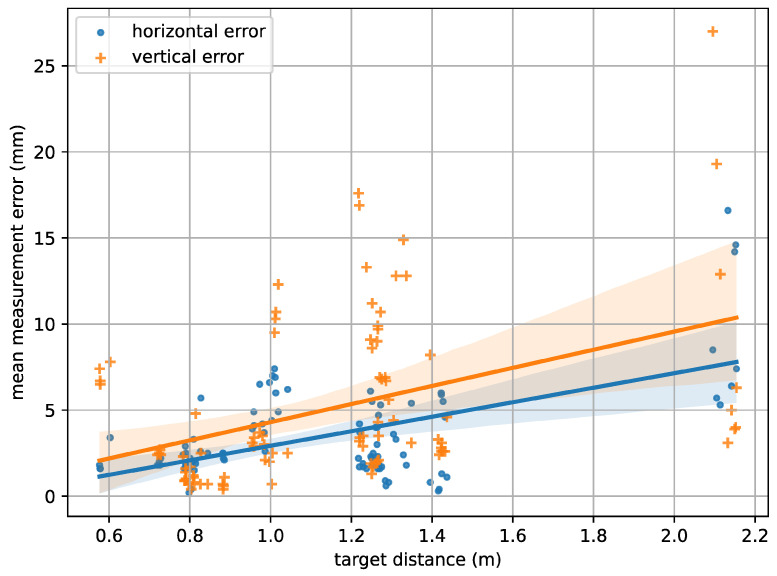
The horizontal and vertical measurement error in relation to the recording distance as has been automatically estimated using a calibration target.

**Figure 7 sensors-23-03311-f007:**
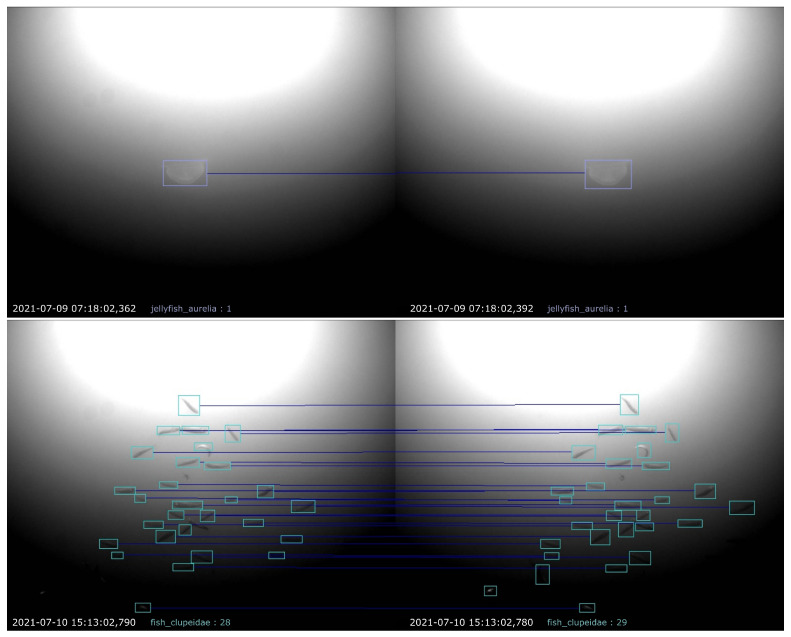
Exemplary results of the proposed matching procedure. Detected animals in both camera views are marked by colored bounding boxes, and boxes matched to each other are represented by horizontal lines drawn between them.

**Figure 8 sensors-23-03311-f008:**
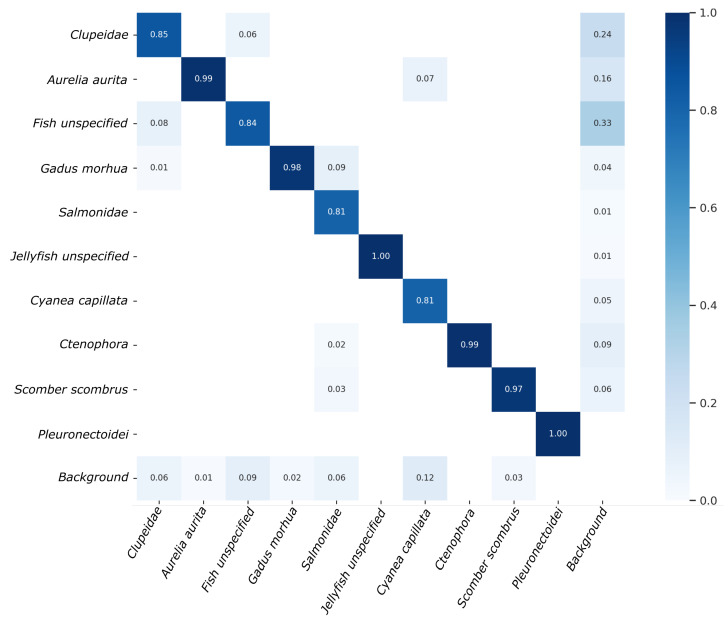
Confusion matrix of the detection results obtained on validation data.

**Figure 9 sensors-23-03311-f009:**
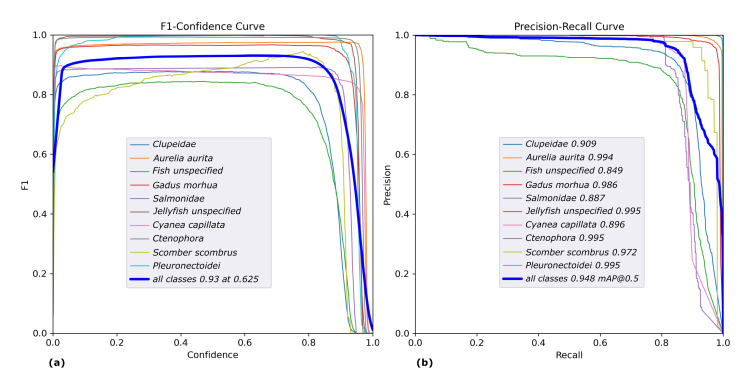
Plot of the class-wise F1 (**a**) and Precision-Recall (**b**) curves of the results obtained on validation data.

**Figure 10 sensors-23-03311-f010:**
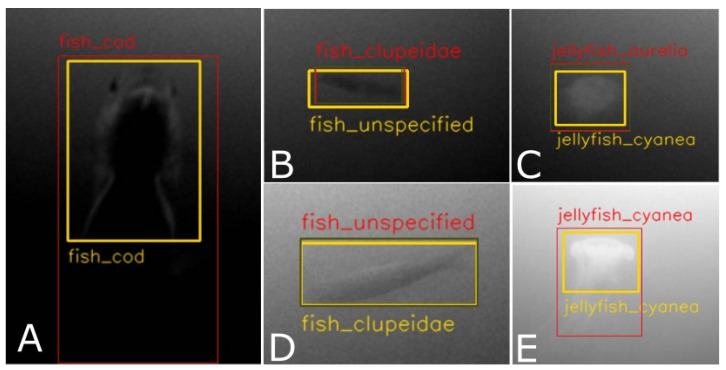
Examples of incorrect detections. The images shown are cropped from the full video frames and are marked with yellow bounding boxes for the target and red bounding boxes for false positives, which either do not match the target with sufficient IoU (0.5) or have incorrect class labels. The examples shown in (**A**,**E**) were rejected due to insufficient IoU, despite correct class labels, while (**B**–**D**) were rejected due to the wrong classes being assigned.

**Figure 11 sensors-23-03311-f011:**
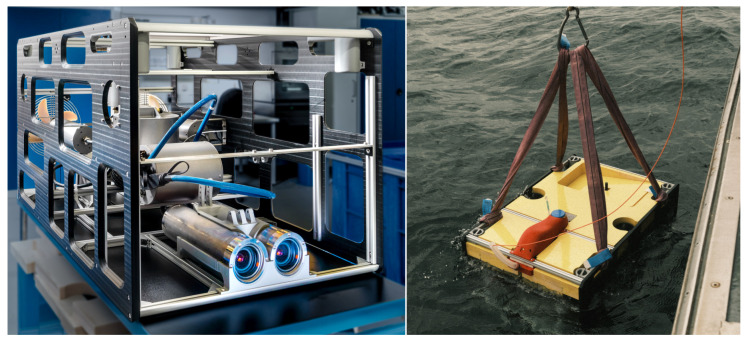
The underwater vehicle for the mobile setup, photographed in the laboratory and during its deployment in the water.

**Table 1 sensors-23-03311-t001:** The amount of annotated images, bounding boxes and tracks for each object class as included in the used dataset.

Species	Images	Bounding Boxes	Tracks
*Ctenophora*	20,695	28,800	203
*Aurelia aurita*	26,542	27,330	109
*Cyanea capillata*	11,838	11,838	26
*Gadus morhua*	7725	8141	210
*Fish unspecified*	3310	6163	215
*Clupeidae*	1846	5001	335
*Jellyfish unspecified*	2160	2262	12
*Salmonidae*	599	1479	23
*Scomber scombrus*	133	964	53
*Pleuronectoidei*	921	921	12
Total	73,144	92,899	1198

**Table 2 sensors-23-03311-t002:** The amount of images and bounding box annotations per category, as contained in the training and validation split.

Species	Images		Bounding Boxes	
	Train	Val	Train	Val
*Ctenophora*	17,412	3283	24,297	4503
*Aurelia aurita*	22,486	4056	23,131	4199
*Cyanea capillata*	9414	2424	9414	2424
*Gadus morhua*	6255	1470	6617	1524
*Fish unspecified*	2564	746	4864	1299
*Clupeidae*	1001	845	3588	1413
*Jellyfish unspecified*	1836	324	1918	344
*Salmonidae*	452	147	1192	287
*Scomber scombrus*	116	17	863	101
*Pleuronectoidei*	774	147	774	147
Total	60,144	13,000	76,658	16,241

## Data Availability

Upon personal request, the data collected in this study will be made available to other researchers who may be interested in further analysis.

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
