# Peer review of "A Deep-Learning Based Pipeline for Estimating the Abundance and Size of Aquatic Organisms in an Unconstrained Underwater Environment from Continuously Captured Stereo Video"

_sensors, 2023, doi:10.3390/s23063311_

Round 1

Reviewer 1 Report

A stationary station, endowed with a stereo rig, has been deployed in the Baltic Sea for eight months, approximately. This manuscript describes materials, methods, and some results from the recorded images. Quite standardized procedures are used to process the images, identify the observed marine species, and quantify the individuals to extract abundance indicators.

In my opinion, the authors are presenting an interesting and sound piece of research, aligned with most innovative current projects devoted to the identification and quantification of marine species. The aim of most of these projects is to obtain environmental indicators and to monitor populations of commercial interest. In addition, the Related Work and Conclusions sections reveal a good knowledge of the research field. I would therefore recommend acceptance of this proposal, provided that the authors correct the following points or provide a reasonable response to my concerns.

In its present form, the title, the abstract and the introduction could mislead the reader, creating expectations about organisms sizing, which in the end turn out to be a bit disappointing because the method used (bounding-box dimension) is extremely simplistic and prone to errors. Moreover, none of the results shown in the manuscript report any measure of this promised parameter. This is my most serious issue regarding the manuscript. So, I would ask the authors to meet the expectation created by the title and the so-called "first contribution of the work" (lines 95&96). Otherwise, I think they should be more modest and focus on those topics for which they actually present results.

Below are some additional comments that I would like you to take into account in order to improve the final version of the manuscript.

Please, add a map of the UFO placement for the recordings in addition to the text in lines 41 to 44. The reference of 45 NM East of Sylt results confusing.

Based on similar experiences, I can assure that making the dataset public could be a much-appreciated contribution to the scientific community.

Although the manuscript is clear and well written, a final proofreading would help to cancel some unusual expressions. See, for example line 113 “Excluded are reports about (…)” instead of “Reports about (…) are excluded”.

Please, provide some run-time data when describing the detection and classification algorithms.

The description of the Matlab calibration algorithm and the Yolov5 features are welcome as they contribute to the completeness of the article, but can be abbreviated, if necessary, to devote more space to other much more particular and original details of the work.

About 50% of the area of the images shown in figure 5 is overexposed. There is no solution to this problem now, but some measure should be considered to correct this problem in future recordings.

Author Response

Dear reviewer,
thank you very much for your valuable comments on our publication. 
We have tried to address all the points mentioned to the best of our ability and think that our report has been significantly improved as a result.

The following listing contains all the changes we've made regarding your comments.

  • Regarding the main criticism about the size measurement -> We agree fully that the designed approach, especially the bounding box matching scheme, is a rather simple approach which has several drawbacks. We also admitted that in the discussion. We have added more content to the publication about the validation of the measured data, i.e. by using the acoustic sonar measurement and calculating concrete error metrics for object size measurements with the calibration target. We believe that this will give a better insight into the performance, which goes beyond looking at the reprojection error alone. However, we are still in the process of performing such validations more extensively for living animals, which is not easy to do since we do not have a ground truth of said animals, besides the sonar data. We agree that the wording in the abstract and in the introduction may paint a too strong picture of the actual size measurement procedure. Therefore, we have added, that the size is but approximated/estimated and also that it is only based on the corner coordinates of the matched bounding boxes, which should emphasize the potential proneness to errors.
  • Please, add a map of the UFO placement for the recordings in addition to the text in lines 41 to 44 -> Updated the figure and added a map visualizing the several deployment locations.
  • About the availability of the data - We absolutely agree, that the sharing of research data is of great importance. As we have detailed in the data availability statement, we will provide the video and annotation data used within this study on the basis of personal requests. Additionally, we are currently also in the process of publishing an extended version of the same database, which will be downloadable from an online resource.
  • Line 113 “Excluded are reports about (…)” instead of “Reports about (…) are excluded” -> Adjusted to the proposed wording
  • Regarding the runtime data -> We have added information about the runtime of the model during inference and for the subsequent non maximum suppression.
  • About the description of the Matlab calibration algorithm and the Yolov5 features -> We have added additional information about the underlying calibration procedure and also elaborated on the difficulties and the systematic measurement error of this method when applied to underwater calibration. Furthermore we explained which steps where taken to reduce the measurement error, e.g. by performing in-situ calibration and capturing the target at many different viewing (measurement) distances. We added a bit more information about yolov5, however would leave the main description of the model to the referenced yolo resources.
  • About the overexposed images -> The overexposure is indeed visible in all shots and is related to the very light sensitive cameras and the integrated automatic exposure control in combination with the strong light gradient from the water surface to the bottom. In contact with the manufacturers, it was not possible to change this behavior using the available camera controls (e.g. changing ROIs used for white balancing). Luckily it was still possible to record the animals in a good quality to perform the classification, although it has happened, that some information is lost due to under or overexposure in the extremal regions of the image. A fact that mitigates this information loss is, that the videos were recorded with a relatively high framerate of 20fps which allows to capture an animal in a large number of shots that in most cases contain at least one good quality image in many cases.

Reviewer 2 Report

See the attached report.

Author Response

Dear reviewer,
thank you very much for your valuable comments on our publication. 
We have tried to address all the points mentioned to the best of our ability and think that our report has been significantly improved as a result.

The following listing contains all the changes we've made regarding your comments.

  • Regarding the literature review -> We agree, that several publications regarding the stereoscopic size measurements were not listed, which mostly is due to the fact, that the publication touches so many different aspects. However, we agree that this part was underrepresented so far, which is why we have added some more of the suggested references and also directed the reader to several reviews that investigate published methods in this widespread field in more detail. 
  • Line 113: Elaborate on the term 'cutouts' -> We have added a more precise description, that regions cropped from full images are meant by this.
  • Line 164: Give R-CNN full term -> We have added the full term for this acronym and added the acronym to the abbreviations section
  • Line 283: Provide one or more references for studies with uneven distributions of data -> Added references to publicly available datasets for marine species detection which also exhibit skewed data distributions (Fish4Knowledge, BrackishMOT). Added a small paragraph about the topic of unbalanced data and added a reference to a review by Johnson et al. for methods to deal with uneven data in Deep Learning. Mentioned the employed data augmentations used in our study to increase data heterogeneity.
  • Line 374: Regarding the validation of the activity detection module -> We expanded the validation of the activity detection on a larger range of sequences from different recording days. However, we did not compute concrete performance metrics, since this step of the pipeline is just a very rough filter to reduce the amount of raw video data. The main part of the pipeline is the subsequent detection module, which is why we didn't provide concrete evaluations of the activity detection, also because in a real-time system this part wouldn't be needed anyway, which has already been described in the paragraph before. Consider this pipeline part to be more of an artifact from the early phase of the project (when no trained model or annotated image data was available). Nonetheless, we have expanded the related paragraph with some additional information about the validation and the characteristics of the activity detection.
  • Line 369: Explain how the ground-truth was determined -> We have included, that the ground-truth is human-annotated. However, the annotation process and thereby the determination of the ground truth has been described in detail before, in subsection "Dataset". Added a reference to the specific subsection in the text.
  • Line 378: What criteria where used to filter out incorrect detections? -> The confidence value is naturally used during the evaluation, e.g. when calculating performance-recall curves. Added another note about this. Also added a short description of the used non maximum suppression which is used to filter out strongly overlapping detection results.
  • Line 382: A figure showing an example of mosaic augmentation would be a useful addition -> In this subsection we aim mainly to list the possibilities of yolov5 e.g. the mosaic augmentations. In fact, while we have used mosaicing in our earlier experiments, we turned it of in our final experiments, as can be seen from the hyperparameters in Appendix table A1. This is also a reason why we didn't add a figure visualizing the augmented images. Nonetheless, we have added an additional paragraph to explain the reasoning why we didn't use mosaic augmentations in the end and added an additional reference to the publication of yolov4 where mosaicing was introduced.
  • Line 396 + Line 398: How was the refraction modeled and where the calibration and measurement distances similar -> Added a more in-depth explanation about the underlying perspective camera model. Also elaborated on the difficulties and the systematic measurement error of this method when applied to underwater calibration. Furthermore explained which steps where taken to reduce the measurement error, e.g. by performing in-situ calibration and capturing the target at many different viewing (measurement) distances.
  • Line 401 What methods were used to independently validate the accuracy of the 3d measurement? -> Indeed, we did not have an independent calibration target like a rod or similar to validate the stereo measurements. While we relied mainly on the reprojection error to assess the performance of the stereo calibration, we also used the synchronized sonar recordings to validate the measurements in some selected cases. We have added additional explanations about this validation and also another figure which illustrates the validation with sonar data for a chosen case. Additionally, we have validated the measurement error using the checkerboard pattern, i.e. by calculating the distance between all vertical and horizontal points of the pattern for all calibration image. We also have added another figure to show how the error relates to the recording distance.
  • Line 435: Provide a reference to the use of MaxN -> Added a reference to an early work of Ellis et al. where MaxN was used to estimate fish abundances
  • Line 442: Provide a reference to the issues associated with re-enumerations in sampling regimes -> Added another paragraph about the difficulties when detecting unmarked animals, and added several references to publications about the topic. However, we did not add more detailed explanations, because we believe this is beyond the scope of this publication
  • Figure 7 -> We have fixed the caption mistake. 
  • Regarding the precision and recall graph -> In our opinion, the precision-recall graph, and especially the computed mAP score, can give a better insight into the models performance as compared to single precision and recall values, because it takes into account both the precision and recall of the model across all levels of confidence thresholds, rather than just a single threshold. Besides the average mAP score over all classes, the score for each single class is also presented in the said figure. Therefore, we suggest that a separate table for single precision recall values is not essential, however would consider adding it nonetheless if it is critical for the approval.

Round 2

Reviewer 1 Report

Thank you for addressing all my comments. Congratulations.